# Insights into the Modification of Carbonous Felt as an Electrode for Vanadium Redox Flow Batteries

**DOI:** 10.3390/ma16103811

**Published:** 2023-05-18

**Authors:** Cong Ding, Zhefei Shen, Ying Zhu, Yuanhui Cheng

**Affiliations:** 1College of Science, Huazhong Agricultural University, Wuhan 430070, China; 2State Key Laboratory of Organic-Inorganic Composites, Beijing University of Chemical Technology, Beijing 100029, China; 3College of Chemical Engineering, Beijing University of Chemical Technology, Beijing 100029, China

**Keywords:** vanadium redox flow battery, electrode, electrocatalysts, surface modification, active sites

## Abstract

The vanadium redox flow battery (VRFB) has been regarded as one of the best potential stationary electrochemical storage systems for its design flexibility, long cycle life, high efficiency, and high safety; it is usually utilized to resolve the fluctuations and intermittent nature of renewable energy sources. As one of the critical components of VRFBs to provide the reaction sites for redox couples, an ideal electrode should possess excellent chemical and electrochemical stability, conductivity, and a low price, as well as good reaction kinetics, hydrophilicity, and electrochemical activity, in order to satisfy the requirements for high-performance VRFBs. However, the most commonly used electrode material, a carbonous felt electrode, such as graphite felt (GF) or carbon felt (CF), suffers from relatively inferior kinetic reversibility and poor catalytic activity toward the V^2+^/V^3+^ and VO^2+^/VO_2_^+^ redox couples, limiting the operation of VRFBs at low current density. Therefore, modified carbon substrates have been extensively investigated to improve vanadium redox reactions. Here, we give a brief review of recent progress in the modification methods of carbonous felt electrodes, such as surface treatment, the deposition of low-cost metal oxides, the doping of nonmetal elements, and complexation with nanostructured carbon materials. Thus, we give new insights into the relationships between the structure and the electrochemical performance, and provide some perspectives for the future development of VRFBs. Through a comprehensive analysis, it is found that the increase in the surface area and active sites are two decisive factors that enhance the performance of carbonous felt electrodes. Based on the varied structural and electrochemical characterizations, the relationship between the surface nature and electrochemical activity, as well as the mechanism of the modified carbon felt electrodes, is also discussed.

## 1. Introduction

With rising concerns over global environmental problems and resource exhaustion, it is crucial to promote the utilization and the development of renewable energy sources, such as solar and wind power. However, the power generated from renewable energy resources such as solar energy and wind energy is intermittent and unstable, which would cause negative impacts if input into the power grid directly. Therefore, it is indispensable to integrate appropriate electrochemical energy storage (EES) devices into the grid to smooth the output of renewable energy production and balance generation and demand according to time and climatic availability [1]. Electrochemical energy storage is a desirable technology that possesses pollution-free operation, high round-trip efficiency, flexible power and energy characteristics to meet different grid functions, a long cycle life, and low maintenance [2]. Specifically, the VRFB is one of the most promising candidate technologies for large-scale energy storage owing to its high safety and reliability, high efficiency, long cycle lifespan, scalability and flexibility of modular design, etc. Therefore, increasing attention has been shown in investigating the VRFB system. Meanwhile, many large-scale demonstrations have emerged in the last few decades, including the world’s largest 5 MW/10 MW h system (in 2012), implemented by Rongke Power Co., Ltd. (Dalian, China). 

In contrast to other flow battery systems, VRFBs store energy by dissolving electro-active species of vanadium ions with different valences in both positive and negative electrolytes. Driven by the circulating pumps, the electrolytes flow cyclically from the outside reservoirs to the electrodes to complete the redox and oxide reactions; meanwhile, chemical energy is converted into electrical energy or vice versa. Figure 1 illustrates the system configuration of the VRFB. The performance of the VRFB is comprehensively affected by the core materials, including the bipolar plate, electrode, electrolyte, and ion exchange membrane (IEM) [3]. The bipolar plate is used as a collecting conduction current. The electrolyte contains vanadium ions and strongly acidic ones such as H_2_SO_4_ are used as the supporting electrolyte. Therefore, the sites that the electrolyte passes through should be resistant to strong acidity. Moreover, the aqueous electrolyte is nonflammable, which prevents the VRFB from ignition and explosion [4]. The ion exchange membrane can conduct protons and prevent the short circuit caused by the direct contact of electrolytes from two half-cells. The electrochemical reactions during charge and discharge are listed as follows:(1)$$\mathrm{Positive}:\;{\mathrm{VO}}^{2+}{+H}_{2}O-e^{-}\overset{\mathrm{charge}}{\underset{\mathrm{discharge}}{\overset{⟶}{⟵}}}{\mathrm{VO}}_{2}{}^{+}{+2H}^{+}{\;\;E}_{0}=1.00\;V$$
(2)$$\mathrm{Negative}:\;V^{3+}{+e}^{-}\overset{\mathrm{charge}}{\underset{\mathrm{discharge}}{\overset{⟶}{⟵}}}V^{2+}{\;\;\;\;\;E}_{0}=-0.26\;V$$

Overall electrochemical reaction: (3)$${\mathrm{VO}}^{2+}+H_{2}{O+V}^{3+}\overset{\mathrm{charge}}{\underset{\mathrm{discharge}}{\overset{⟶}{⟵}}}{\mathrm{VO}}_{2}{}^{+}{+V}^{2+}{+2H}^{+}{\;E}_{0}=1.26\;V$$

Among the VRFB components, the electrode is the predominant factor that constrains VRFB performance. The electrode provides the transfer channel and active sites for the electrochemical reaction; therefore, the inner structure is of profound importance in improving the catalytic activity as well as the electrochemical reversibility of the electrode and even augmenting the energy and power densities of the VRFBs.

Various metals, such as gold, lead, titanium, platinum, and titanium-based platinum, had been employed as electrodes for VRFBs in the past due to their good conductivity and malleability [5]. Nevertheless, the noble metals are expensive, have inferior reversibility, show the formation of a passivation layer, and are susceptible to hydrogen evolution. Thus, they were gradually eliminated by the market. With low costs and high chemical stability, carbonous felt electrodes are more suitable for VRFBs. There are many types of carbonaceous materials, but only typical types have been commonly used in recent years, including graphite felt (GF), carbon felt (CF), carbon paper (CP), carbon cloth (CC), graphene oxide (GO), nanoscale materials such as carbon nanotubes (CNTs), carbon nanospheres (CNS), carbon nanofibers (CNFs), and carbon nanoparticles (CNPs), etc. [6,7]. Through extensive investigation, CF and GF, two commercial carbonous felt electrodes, are selected as the best candidates for substrate electrodes for VRFBs because of their uniformly three-dimensional (3D) structure, specific surface area, and wide operating potential window. However, there is still an urgent need to modify carbonous felt to resolve the poor electrochemical activity and reversibility, which is ascribed to the hydrophobic surface nature and poor kinetics involved in the conversion between different vanadium redox-active species, especially the more sluggish positive reaction between VO^2+^/VO_2_^+^ [8]. However, there is still a discussion regarding whether VO^2+^/VO_2_^+^ or V^2+^/V^3+^ limits the performance of VRFBs, so some researchers focus on the reaction kinetics of the cathode [9]. In other words, reducing the electrochemical and ohmic polarization is essential to realize the high energy efficiency (EE) of a battery [10]. Moreover, reducing the cell resistivity will increase the power density, which in turn reduces the VRFB cost per kW [11]. In this regard, there has been a continuous effort to enhance the activity of carbonous felt in the past few decades. Various modifications to the carbonous felt surface have been carried out to resolve these problems, which can be classified into four categories: surface treatment or structure rebuilding, metal or metal oxide modification, nonmetal modification, and carbonous nanomaterial decoration. 

In this review, we also focus on the technical trends in each modification method category, including the preparation and decoration of carbon-based felt electrodes, structure and morphology characterization, and electrochemical or cell performance evaluation for the VRFB. After taking the reagent and method for comparison, we determine the factors influencing the performance of a VRFB with a modified electrode. Moreover, by listing all modified electrodes in the same type together, we attempt to determine the relationship between the surface nature and electrochemical activity, as well as the mechanism of the modified carbon felt electrode. The main objective of this paper is to overview the progress made in the modification of carbonous felt electrodes in the last few decades to clarify the state-of-the-art developments and the trends from a different perspective.

## 2. Literature Review of Modification of Carbonous Felt

Since redox reaction kinetics and concentration polarization strongly rely on the morphology and surface chemistry of electrodes, modifications to the electrode’s surface are a significant approach to raising the electrochemical performance of batteries by alleviating polarization. Intensive studies have attempted to modify carbonous felt’s surface chemistry through physical or chemical treatments. For instance, plasma, thermal, and chemical treatments are efficient approaches to change the hydrophobic surfaces of bare felts to hydrophilic. Generally, there are two widely applied approaches to prepare high-performance GF electrodes. The first one is treating the fiber surface of the carbonous felt with an oxidant and physical or chemical methods, generally known as surface treatment. The other one is to etch the GF surface to boost the oxygen functional groups and effective surface areas.

### 2.1. Surface Treatment or Structure Rebuilding

The VRFB electrode modification process has a long history, since Prof. M. Skyllas-Kazacos first proposed the intrinsic modification of carbonous felt materials by thermal treatment [12] and chemical treatment [13] in 1992. The researchers concluded that the high content of surface oxygen-containing functional groups, such as hydroxyl (–OH) and carboxyl (–COOH) groups, considerably enhanced the activity of the GF electrode for the vanadium redox reaction compared with untreated samples. Oxygen defects on the GF surface displayed both catalytic effects in vanadium redox reactions and good wettability by aqueous electrolytes, thus leading to reduced polarization. From then on, various facile methods to apply oxygen functionalization to the surfaces of electrodes have been widely utilized, including microwave [14], square wave potential pulse [15], plasma [16], hydrothermal [17], and thermal treatment [18,19,20,21,22,23,24,25,26,27], by introducing oxidants such as acids [17], bases [28], oxygen [14,18,19,20,26,27], oxygen plasma [16], H_2_O_2_ [16,17], KMnO_4_ [24], and K_2_CrO_4_ [29]. 

The microwave-assisted method is a simple and efficient method for electrode modification with CF or GF; it utilizes microwave irradiation to heat carbon substrates to a high temperature of several hundred degrees Celsius within a few minutes or even seconds. For example, Cho et al. [14] modified CF with 650 W microwave treatment in a mixture of O_2_ and Ar for 1 min to gain a more roughened surface. Then, both the specific surface area and the amounts of oxygen functional groups of the treated CF rose. The aim of O_2_ plasma treatment is to ameliorate the wettability of commercial felt by introducing oxygen-containing functional groups to the exterior of fibers to enhance the electrochemical reactivity. Estevez et al. [16] treated GF with O_2_ plasma and subsequent oxidation with H_2_O_2_ to enrich the oxygen functionalized active sites, thus improving the EE by almost 8% at 150 mA·cm^−2^ compared to thermally treated GF in air. 

Moreover, many studies have attempted to modify the surface functionality of the carbonaceous electrode via wet chemical methods such as hydrothermal treatment and thermal treatment. He et al. [17] treated GF with HF and then hydrothermally treated it with H_2_O_2_. The EE of the treated electrode was enhanced by 4.1% at 50 mA·cm^−2^. Besides H_2_O_2_, KMnO_4_ is also a typical strong oxidant. For instance, Jiang et al. [30] treated GF with KMnO_4_ solution at room temperature for 5 days. The EE reached 84.0% at 200 mA·cm^−2^ and the structure of the electrode remained stable and intact over more than 500 cycles. Modification methods including ultrasonic and microwave are all physical means, which promote sufficient contact between the oxidants and electrode. The main purpose of these methods is to enlarge the surface area or promote the quantity of active sites by introducing more functional groups onto the material’s surface, such as –OH, carbonyl (–CO), and –COOH, which facilitates electron transfer and thus reduces the overpotential. However, these methods are unsuitable for practical application when using strong oxidants and performing difficult and time-consuming processing steps. 

Similar to the effect of plasma treatment, facile thermal treatment is also an efficient method to prepare robust and plentiful oxygen-containing functional groups by altering the surroundings of the carbonous electrode with different gaseous atmospheres, such as air [18,19], CO_2_ [20], or water vapor [21]. For example, Kim et al. [18] heated the CF at 500 °C in the air for 5 h, and they also treated the CF with oxygen plasma and gamma-ray for comparison. The battery assembled with the air-modified CF showed high stability over 500 cycles and a high EE of 75% at 40 mA·cm^−2^. Similarly, as reported by Park et al. [19], CF treated in air by heating at 450 °C for 5 h showed an increase in the EE from 65% to 73% at 50 mA·cm^−2^. Instead of oxidizing the electrode with O_2_, Chang et al. [20] heated GF in a CO_2_ flow at 1000 °C for 30 min to create defects and increase the oxygen functional groups. The EE of the modified GF increased to 84.15% at 50 mA·cm^−2^, which was higher than that of the air-treated CF. Although the etching of CO_2_ shows superiority over air treatment, the treatment temperature and cost are too high. Moreover, Kabtamu et al. [21] etched GF with water vapor injection for 5 min at 700 °C to create more uniform small holes, a rougher surface area, and higher content of oxygen-containing functional groups, and the electrode displayed EE of 83.1% at 50 mA·cm^−2^. In conclusion, although the surroundings of the thermally treated electrode differed, these studies utilized a common mechanism to increase the oxygen functional groups and effective surface areas by etching the GF surface. 

Although some attempts have been made to promote the catalytic activity of carbon-based electrodes, the specific surface area is still inadequate and the stability of the electrode structure over a long period of battery cycling is difficult to maintain. Therefore, electrodes with a nanohole structure, prepared by a simple, controllable, and cost-effective method, namely chemical etching, have emerged. Nanomaterials such as NiO [22], FeOOH [23], Co_3_O_4_ [31], CuO [32], Fe_2_(SO_4_)_3_ [33], and K_2_FeO_4_ [34] can be grown on the surface of the electrode by immersing the electrode in a precursor solution (such as (NO_3_)_2_·6H_2_O and FeOOH, etc.), and then large numbers of uniform pores would be generated by dissolving and removing the metal oxide nanoparticles with acid. For instance, Park et al. [22] immersed GF in a precursor (Ni(NO_3_)_2_·6H_2_O) solution in acetone, followed by the repetition of the NiO/Ni redox reactions through a stepwise thermal annealing process to generate desirable porous structures and plentiful oxygen defects on the stepped edges of the optimized material. Then, Liu et al. [23] used a similar three-step method to create defects by growing FeOOH nanorods on thermally treated GF via a hydrothermal method at 95 °C initially, and subsequently with thermal reduction in N_2_ at 900 °C to generate Fe_3_O_4_, which was then removed by acid later. Finally, only nanoholes remained on the GF, without any metal residues. Furthermore, the size as well as the depth of pores could be flexibly tailored, showing enhanced wettability and outstanding cycle stability (as shown in Figure 2).

Although many breakthroughs have been made, previous electrodes were generally prepared at temperatures higher than 500 °C and the presence of a N_2_/Ar atmosphere is indispensable, raising the cost of preparation and limiting their large-scale production. The relatively low performance of the GF electrodes has also hindered further cost reductions for VRFBs. Aiming to realizing energy conservation, novel catalytic etching methods have been investigated. For example, Abbas et al. [31] immersed CF in cobalt acetate tetrahydrate Co(Mn(CH_3_COO)_2_·4H_2_O) to form a layer of metal precursor, which was then oxidized to Co_3_O_4_ at a low temperature of 300 °C and at a range of 300~500 °C to create holes in the final product (Figure 3). Moreover, Wang et al. [34] chose K_2_FeO_4_ to etch the GF via thermal treatment for the generation of nanopores of about 20 nm and mesopores of about 0.5 μm. Though working at a high current density of 500 mA·cm^−2^, the VRFB with gradient-pore-oriented GF still exhibited excellent performance, with EE of 63.41%. Similarly, Jiang et al. [24] heated GF previously immersed in FeCl_3_ solution and then washed it with HCl to remove the residuals, thus forming a bi-porous structure consisting of ~100 μm primary pores and ~200 nm secondary pores, acting as the macroscopic transfer pathways for the flowing electrolyte and increasing the active surfaces, respectively. The bi-porous electrode exhibited notable EE of 82.47% at 300 mA·cm^−2^, among the best performance in the published literature at that time. Nevertheless, the metal salts applied for the growth of nanomaterials in these studies were etched finally using a strong acid. Moreover, inferior electrical conductivity inhibited the charge transfer process on the electrode. Thus, a highly conductive electrocatalyst support is needed to resolve this problem.

On the whole, treating an electrode with different methods and oxidants, heating the electrode in different gas atmospheres, and forming nanoholes by eliminating nanomaterials are the three predominant methods to intrinsically ameliorate the surface or even change the structure of the electrode. In addition, the aims of these studies are almost similar, which is to increase the specific surface area and determine the amount of oxygen functional groups on the electrode surfaces, ensuring that the GF has a more hydrophilic interface and sufficient transportation channels for active redox couples. Surface modification has become significant in maximizing the specific surface areas of carbon materials, while also ensuring excellent electrical conductivity. Moreover, the generated functional groups are also thought to serve as reactive sites for the redox reactions of vanadium ions. 

Most experiments show the catalytic performance of modified carbon-based materials and pristine materials. However, due to the different experimental conditions (area of electrode, current density, etc.), a comparison between many experiments becomes challenging. It is preferred to establish standard experimental procedures or experimental parameters for VRFBs’ electrocatalytic performance. To improve our knowledge of the performance of surface-treated or structure-rebuilt carbonous felt, we list the detailed cell performance of the electrodes mentioned in this section in Table 1, considering the sizes of the modified electrodes used and the highest current density.

To compare the performance of modified electrodes more comprehensively, we summarize the previous works performed with various electrodes modified by intrinsic treatment methods aforementioned for VRFBs (as given in Table 1). By means of various surface treatments, the charge–discharge current density of the VRFB increases from the initial 50–150 mA·cm^−2^ to the current 100–300 mA·cm^−2^. As we know, to satisfy commercial applications in industry, the current density and the power density of VRFBs are crucial to reduce the installed costs of VRFBs by decreasing the material consumption and stack size. However, a higher current density will lead to polarization loss and lower EE in a VRFB. Therefore, the EE of a VRFB under high current density is crucial to evaluate the polarization increase and electrochemical performance of electrode materials. As illustrated in Table 1, the highest reported current density almost shows a constantly rising trend, ranging from 30 to 300 mA·cm^−2^, reflecting the ambition of most researchers to achieve optimized electrode materials with higher battery performance. As for the structure differences, the specific surface area, and the content of oxygen functional groups caused by the diverse physical or chemical treatment methods, EE shows obvious diversity. More importantly, the inconsistent area size of the electrode also makes a difference. Some researchers have considered that the excessive surface oxidation of carbon materials will lead to the corrosion of the electrode [36]. Moreover, the oxygen functional groups may not be stable enough during cycling, leading to the loss of oxygen content and deteriorated performance [37]. Although the rate capability of VRFBs is substantially enhanced, the long-term cycle stability of the modified GF still remains to be improved due to the harsh acidic environment and the continuous flushing of the oxidizing electrolyte. 

### 2.2. Metallic Modification

Metals or metal oxides are usually employed as electrochemical catalysts to modify carbonous felt electrodes at low abundance, which is entirely different from the concept of metal electrodes mentioned in the Introduction. In consideration of the cost, precursors containing cheap metals are attractive and have been investigated extensively in recent years. To enhance the reactivity toward the redox reactions of VRFBs, Skyllas-Kazacos et al. [38] initially performed metal incorporation (e.g., Pt, Ir, Mn, and In) into the carbon fiber electrode via a wet chemical method. Since then, many researchers have attempted doping with various metals to accelerate the kinetics of electrode reactions. For instance, Wei et al. [39] coated GF with uniformly distributed copper nanoparticles by electrodeposition, displaying a significant enhancement in the electrochemical kinetics of the V^2+^/V^3+^ redox reaction. The EE of the Cu-coated electrode increased by 17.8% over the pristine GF at 300 mA·cm^−2^. However, the mechanism of Cu modification has not been clarified and needs to be studied further. 

Afterwards, researchers found that Au, Mn, Pt, Ir, Ru, Os, Re, Rh, Sb, Te, Pb, and Ag have potential for incorporation into carbonous felt electrodes to enhance the positive electrode reaction kinetics. Moreover, Pb, Bi, Tl, Hg, Cd, In, Ag, Be, Ga, Sb, As, Zn, Ca, and/or Mg, etc., impregnated with carbonous felt electrodes, could inhibit hydrogen evolution at the negative half-cell during the charging process, which is one of the main factors causing charge imbalance and undesirable efficiency losses in the VRFB. Among all these electrocatalysts, bismuth is a controversial candidate discussed in many studies [40]. Gonzalez et al. [41] decorated GF with bismuth, a single main group metal, by heating at 450 °C for 4 h in air. The Bi-NP-coated GF had enhanced performance toward VO^2+^/VO_2_^+^ at low metal content but the performance of the VRFB was not mentioned. Meanwhile, Li et al. [42] attached Bi NPs onto GF by synchronous electrodeposition with Bi^3+^ in both positive and negative electrolytes. However, the Bi NPs were only found in the negative side due to the redox potential of Bi/Bi^3+^. Both of these studies modified Bi NPs on GF, but showed different results in terms of cyclic voltammetry. It is necessary to confirm whether the difference was caused by the methods. Moreover, Bi NPs modified on GF by Suarez et al. [43] also enhanced the performance of the negative reaction, and they confirmed the formation of BiH_x_, restraining the competitive irreversible reaction of hydrogen formation. Among these metals, bismuth and its oxides are more suitable for the anodic reaction because of their excellent inhibition of hydrogen evolution.

#### 2.2.1. Single Metal Oxide Doping 

Although precious metals such as Au, Ir, Pt, Pd, and Ru, deposited as catalysts on the surface of carbonous felt, showed high catalytic activity for the VO^2+^/VO_2_^+^ couple reaction and seemed to be promising in achieving the better performance of the VRFB, they are not applicable due to the high costs and susceptibility to water molecule splitting. As an alternative approach to enhancing the catalytic activity of GF, many efforts have been devoted to discovering rationally designed metal oxide catalysts with a lower price and depositing them on the GF surface to enhance the electrocatalytic activity of GF. 

To enhance the kinetics of VO^2+^/VO_2_^+^, low-cost metal oxides have been investigated as alternative electrocatalysts for redox reactions in succession, such as Nd_2_O_3_ [44], Ta_2_O_5_, [45], SnO_2_ [46], WO_3_ [47,48], MoO_2_ [49], CeO_2_ [50], ZrO_2_ [51], PbO_2_ [52], MnO_2_ [53], Mn_3_O_4_ [54,55], Cr_2_O_3_ [56], CoO [57], NiO [58], and Nb_2_O_5_ [59]. The growth of metal materials on the surfaces of felt electrodes is conducted via treatments such as electrodeposition, impregnation, and thermal reduction, etc. The first approach is to electrodeposit metal onto the surface of the carbon material by means of a galvanic bath under a given current. The second type is to immerse the carbon material in a solution containing the metallic ions and then thermally treat it in air at high temperatures. The amount of metal on the fiber surface could be adjusted varied depending on the modification method and the treatment time.

According to the report of M. Skyllas-Kazacos [60], PbO_2_ deposited on a positive or negative electrode during cycling could activate the reaction or inhibit the evolution of H_2_, respectively. Inspired by this work, Wu et al. [52] modified PbO_2_ composed of a mixture of orthorhombic α-PbO_2_ and tetragonal β-PbO_2_ crystallographic phases onto GF in the positive electrode by pulse electrodeposition to form a dense layer of PbO_2_ (Figure 4), and the as-prepared electrode showed good chemical stability in corrosive media and a high overpotential value for the oxygen evolution reaction, reducing the charge transfer resistance and exhibiting EE of 82.0% at 70 mA·cm^−2^.

To further enlarge the current density of the VRFB operation, Xiang et al. [57] proposed the facile metal oxide modification of GF via impregnation combined with ultra-sonication, followed by calcination to uniformly coat a rough, ultra-thin layer of metal oxide on the surface of the carbon fiber (Figure 5). They then doped GF with Cr_2_O_3_ [56] and CoO [57], respectively, both under thermal treatment in air at 400 °C for 2 h and in N_2_ at 500 °C for 5 h. The Cr_2_O_3_- or CoO-modified GF exhibited an increase in the EE by 10.8% or 12.7% at 150 mA·cm^−2^ over the pristine GF, respectively. Notably, the discharge capacity of CoO-modified GF, an important parameter reflecting the electrolyte utilization depth, showed a remarkable enhancement of 101.7% over the pristine GF, ascribed to the faster diffusion process and reduced electrochemical polarization.

NiO, with a low cost, is an important semiconducting oxide with a wide band gap [61], showing benefits toward the kinetics of V^2+^/V^3+^ and VO^2+^/VO_2_^+^ redox reactions. For instance, Yun et al. [58] modified GF with NiO nanoparticles to achieve an increase in oxygen groups and exhibited EE of 74.5% at 125 mA·cm^−2^.

The uniformity of the modified material influences the performance of the electrode greatly, so hydrothermal or solvothermal methods are normally applied in modifying carbonous felt to effectively generate uniform, nanosized materials with a high specific surface area and finally enhance the performance of the electrode. For instance, Ta_2_O_5_ prepared by solvothermal treatment at 240 °C for 12 h [45] showed better nanosized crystallinity (Figure 6), compared with that prepared by sol−gel and hot filament metal vapor deposition methods [62]. Moreover, the uniformly immobilized Ta_2_O_5_ nanoparticles modified the GF with a higher surface area and enhanced hydrophilicity and catalytic activity; it exhibited an increase in EE by 9% and stability after 100 cycles at 80 mA·cm^−2^ over the original GF. The enhancement in performance may be ascribed to the presence of four Ta=O and two Ta-O bonds in tantalum oxide (Ta_2_O_5_), which might be the critical factor for the notable improvement in the electrochemical activity of the Ta_2_O_5_-GF nanocomposite electrode. Furthermore, as a type of single main group metal oxide, SnO_2_ shows stability in sulfuric acid electrolyte solution. Therefore, Mehboob et al. [46] decorated SnO_2_ nanoparticles onto CF via a hydrothermal method at 150 °C for 4 h. The VRFB with SnO_2_-treated CF exhibited EE of 77.3% at 150 mA·cm^−2^ compared with pristine GF. Even at a lower temperature than in Batyeh’s work [45], the outstanding performance indicated the superiority of SnO_2_ modification of GF.

Among numerous transition metal oxides, tungsten oxides are typical and effective electrocatalysts that have attracted extensive interest owing to their resistance to corrosion in acidic solutions, nontoxicity, and the unique chemical- and size-dependent properties of the 1D structure. Pure WO_3_ or WO_2_ shows low conductivity and a limited number of active sites [47,48], while nanosized tungsten oxides in a substoichiometric composition formed with an equal defect structure show a high surface area and active sites. Modified GF with W_18_O_49_ nanowires (W_18_O_49_NWs) prepared via the solvothermal method displayed enhanced EE by 12.5% over pristine GF at 80 mA·cm^−2^ due to the increase in oxygen vacancies (Figure 7) [63]. As the literature shows, the catalytic effect may be improved by the feasible adsorption of VO^2+^ and the exchange of ions on the surface of the electrode, which are functionalized with W_18_O_49_NWs, and the faster electron-transfer reaction from VO^2+^ to form VO_2_^+^. In summary, hydrothermal or solvothermal methods seem to be more suitable compared with routine thermal methods in the terms of preparation methods for the doping of uniformly dispersed, nanosized metal oxides on GF.

#### 2.2.2. Transition Metal Carbides, Nitrides, and Carbonitrides 

To enhance the kinetics of V^2+^/V^3+^, TiO_2_ [64,65] and its derivatives [66,67] have been investigated as alternative electrocatalysts for VRFBs. TiO_2_ dramatically reduces the Hydrogen Evolution Reaction (HER) rate by several orders of magnitude, but it showed little effect on the performance of VRFBs. Later, Vazquez-Galvan et al. [64] used titanium butoxide to modify GF by hydrothermal treatment, followed by annealing to directly grow TiO_2_:H nanorods on the surfaces of graphite felts to introduce several defects in the crystal structure and improve the electrochemical activity in the charge transfer. Moreover, they replaced the mixture of H_2_ and Ar with NH_3_ in another work [65]. The NH_3_-treated TiO_2_ introduced N-containing functional groups and showed EE of 71% at 150 mA·cm^−2^. 

Furthermore, transition metal carbides, nitrides, and carbonitrides have been attracting increasing attention due to their remarkable properties, such as a high melting point, exceptional chemical stability, and high hardness [68]. Wei et al. [66] decorated GF with TiO_2_ nanowires by hydrothermal treatment and then transferred it to TiN under a treatment with NH_3_. The treated electrode showed an improvement in electrolyte utilization by 43.3% and the EE increased by 15.4% at 300 mA·cm^−2^ over the original one (Figure 8). With regard to titanium carbides, Ghimire et al. [67] chose TiF_4_ as a precursor to decorate GF with TiO_2_ particles by hydrothermal treatment and transferred it to TiC by carbothermal treatment at 1250 °C in Ar; it finally exhibited an increase in EE by 13% at 100 mA·cm^−2^.

MXene is a typical transition metal carbide and nitride [69], and Wei et al. [70] synthesized Ti_3_C_2_T_x_ and decorated it onto GF. The EE of a VRFB assembled with Ti_3_C_2_T_x_-decorated GF was 75% at 300 mA·cm^−2^. The as-prepared hollow Ti_3_C_2_T_x_ spheres in the form of 3D-nanostructured MXene could decorate carbonous felt uniformly and showed superiority as an electrocatalyst. Similar to Table 1, the cell performance and corresponding current density of the electrodes mentioned above are summarized in Table 2.

The purpose of metallic modification is to improve the electrical conductivity of felt electrode materials, ensuring lower polarization and better kinetic reversibility toward vanadium redox reactions. The enhancement in the performance of metal-oxide-modified electrodes may be also caused by the abundant oxygen functional groups generated from the metal oxide. Carbonous electrodes decorated with metallic electrocatalysts exhibit higher battery performance and seem to be more economical compared with the precious metal ones. Nevertheless, most of these procedures are still highly energy-/time-consuming and require the use of environmentally unfriendly and expensive chemical reagents, which is undesirable for the synthesis of large-area electrodes for large-scale energy storage applications. The cost of electrode modification must be further reduced to satisfy the demands of the practical application of VRFBs. Furthermore, their stability and the degradation in battery lifetime should not be neglected. As a vanadium electrolyte with high acidity constantly flows and scours the electrode surface, these grown metal species, including metal oxides, with weak adhesion may fall away from the surface of the matrix during the long cycling process and eventually lose their electrocatalyst effect. Therefore, it is crucial to develop other electrode materials with lower costs and better stability, such as nanosized metal alloys with strong corrosion resistance, a large surface area, and high electrical conductivity.

### 2.3. Nonmetal Doping

Although the rate capability of VRFBs is substantially enhanced through the previous two modification methods, the long-term cycle stability of the modified GF is still insufficient. Herein, other strategies are put forward to solve the cost and durability problems.

It is well accepted that the doping of heteroatoms contributes to the variations in the electron distribution and local bonding environment on the surfaces of carbon-based materials, resulting in the outstanding performance of the electrodes or electrocatalysts in fuel cells, lithium-ion batteries, etc. [75]. Inspired by these notions, Shao et al. [76] applied nitrogen doping for VRFB electrode modification to promote the electrochemical properties of the VO^2+^/VO_2_^+^ redox couple for VRFBs. Since then, besides nitrogen [77,78,79,80,81,82,83,84,85,86], doping with other heteroatoms, such as phosphorus [87], halogen, and boron [88,89,90,91], as well as dual doping with carbon–nitrogen [92,93,94,95], phosphorus-doped C_3_N_4_ [96], oxygen–nitrogen [97,98,99,100,101], oxygen–phosphorus [102,103], fluorine–phosphorus [104], nitrogen–phosphorus [105], and nitrogen–sulfur [106], have been applied in VRFBs and demonstrated excellent rate performance and outstanding cycling stability.

In particular, nitrogen has been widely regarded as a typical dopant for numerous carbon materials for EES, because it promotes the charge transfer of the electrode/electrolyte interface by abating the activation energy for the redox reaction and by offering supplementary active sites [107,108]. Recent studies have also reported several types of nitrogen-doped carbon electrodes, obtained generally by the impregnation and chemical vapor deposition (CVD) of various N-containing hydrocarbons [98], providing considerable increments in the VO^2+^/VO_2_^+^ redox kinetics [47,109,110]. Organics are normally chosen as heteroatom sources, while inorganics such as NH_3_ or even N_2_ plasma are also considered as efficient N sources in modification, aiming to increase the wettability of commercial felt electrodes and boost the electrochemical reactivity. Compared with the commonly employed thermal activation process, the plasma treatment process is efficient, and subsequent physical or chemical changes incurred by plasma treatment will be uniform across the felt, while the surface area remains intact. 

Much of this work has shown that various forms of amine groups can be used as efficient surface modifiers to act as active sites for vanadium redox reactions. As commercialized N-doped carbon powders block the passageway for the electrolyte in GF, typical organics such as pyrrole and dopamine, serving as efficient nitrogen sources, could be coated on the GF by polymerization and pyrolysis. For instance, Park et al. [77] modified GF with nitrogen by simply heating it at 900 °C, causing the formation of a polypyrrole layer with the assistance of Co. The EE of the cell with nitrogen-treated GF was 13.8% higher than for the pristine one at 150 mA·cm^−2^. Meanwhile, Lee et al. [78] and Youn et al. [79] grew a layer of polydopamine (PDA) on GF and the EE of the cell with the modified electrode was 12.6% or 6.3% higher than that of the pristine one, respectively.

As the structural stability of either carbonous felt or a catalyst is of great importance in modification, researchers have considered several methods to coat carbonous felt with catalysts composed of N and C simultaneously. For example, Schnucklake et al. [111] chose 1-butyl-3-methyl-pyridinium dicyanamide as the source of N and C to modify CF, combined with the salt-templating of sodium chloride and zinc chloride, under the catalysis of iron. Moreover, to form pyridinic-N and maintain the natural structure of the GF, Ma et al. [92] directly formed a coating of carbon nitrides on it by thermal treatment with urea. The EE of the modified GF was about 75% at 150 mA·cm^−2^ after 800 cycles. While the carbon nitride coated on the GF was out of order, the ion-electron transport network was not formed successfully. Therefore, Sheng et al. [93] decorated GF with Zn(NO_3_)_2_·6H_2_O and 2-methylimidazole and then modified it with a pyrolysis process in Ar to obtain an N-doped carbon-film-bridged framework. The EE of the carbon-nitride-decorated GF was 74.3% at 200 mA·cm^−2^ (Figure 9). Such high performance indicates that the structure of the catalyst also needs to be taken into account.

Zhang et al. [94] decorated GF with a polymer layer of p-phenylenediamine and phytic acid by hydrothermal treatment at 100 °C for 10 h and then heated it at 900 °C for 1 h in an inert atmosphere. The N-doped carbon network electrode had a decrease in peak potential by 357 mV and an increase in the EE by 6% at 200 mA·cm^−2^. It is well accepted that the introduction of N atoms into the carbon framework can create defects and vacancies in the carbon matrix and alter the charge distribution between C and N atoms, benefitting the vanadium ions’ mobility and electronic conductivity. Furthermore, free electrons can be ionized out when the electronegative N atoms are introduced into the carbon framework, being conducive to ion exchange and the redox reaction. Yang et al. [95] decorated a robust and stable polymer of intrinsic microporosity (PIM) onto GF and the N-doped multimodal porous carbon exhibited adequate active catalytic sites and efficient ion and electron transport pathways, showing enhanced electrochemical reactivity and wettability.

Moreover, the co-doping of heteroatoms can further promote the properties of materials due to the larger amount of heteroatoms and the synergistic effect of co-doped heteroatoms [112,113,114]. Co-functionalization of carbon electrodes by N and O surface groups markedly facilitated the VO^2+^/VO_2_^+^ or V^2+^/V^3+^ redox kinetics in VRFBs. With the introduction of N functional groups or increasing O functional groups, cooperation between N and O may benefit the performance of carbonous felt according to the modification of binary metal oxides. Thus, Kim et al. [98] used urea as an N and O source to modify GF. In contrast, melamine was chosen as an N source for modification. Meanwhile, they modified GF with NH_3_-O_2_ by CVD. Both routes increased the amounts of N and O functional groups on the GF with simple reagents. The increase in the EE was only about 3% to 4% at 80 mA·cm^−2^, although the initial charge–discharge capacity was almost 1.6 times higher than that of the conventional oxygen-doped GF electrode. Moreover, they [97] also treated GF by heating it with a mixture of NH_3_ and O_2_. The voltage and EE of the cell with N and O co-doped GF were enhanced by about 4–6% at 110 mA·cm^−2^ compared with the heated GF and were also higher than those of N or O single-doped GF. 

With the same outer electron structure as nitrogen, phosphorous has potential to be a good catalyst candidate in carbonous felt modification. Phosphorus atom doping could disrupt the stable π-bond conjugation system between graphite carbon atoms through the larger atomic radius of the phosphorus atom. P-doped electrodes promote the catalytic activity of the vanadium ion redox reaction by introducing defect sites and changing the electron cloud distribution around adjacent carbon atoms. For instance, Yu et al. [87] placed GF into hydroxyethylidene diphosphonic acid (HEDP) by hydrothermal treatment. Then, the cell fabricated with the prepared electrode PGF was used at an extremely high current density of 400 mA·cm^−2^ and exhibited stable EE of 79% at 150 mA·cm^−2^ during 1000 cycles. With excellent rate capability and cycling stability, the PGF also showed good durability under harsh conditions, making it a promising electrode for the next-generation VRFB operating in high-power and all-climate conditions. 

Besides organics, inorganic salt is also an alternative heteroatom source with which to introduce phosphorous functional groups onto carbonous felt. For instance, Kim et al. [102] treated GF via the thermal decomposition of NH_4_PF_6_ in a pyrolysis process. Then, oxygen-rich phosphate groups could be successfully integrated into the surface of the CF, exhibiting good hydrophilicity. Assembled with the treated GF, the cell showed EE of 88.2% at 32 mA·cm^−2^. The most common form of P and O was phosphate radicals, so they deemed that P worked as a bridge between hydrogen and carbon, promoting the process of electron transfer. Similarly, Huang et al. [104] doped GF with P and F functional groups by thermal treatment with KPF_6_, thus raising the number of oxygen-containing groups and bringing in plentiful defects. The heteroatom co-doped electrode showed outstanding reaction activity and EE of 54.9% at 250 mA·cm^−2^, displaying remarkable stability and durability after 1000 cycles. Above all, the covalent bond between heteroatoms and carbon is proven to extend the electrode’s operating life. To reveal the internal structural variations in carbon electrode materials, Wu et al. [103] heated GF with KH_2_PO_4_ to prepare a carbon fiber prototype system with a gradient distribution of phosphorous-containing functional groups (Figure 10) and clarified the mechanism of the enhanced redox reaction kinetics with the assistance of density functional theory (DFT) calculations. They verified that the gradient distribution of P functional groups with a narrow HOMO–LUMO energy gap and low adsorption energy provided the modified GF with excellent redox reversibility and prominent power capability.

Different from the cooperation of N and O, the combination of N and P has the same outer electron structure but the electronegativity is distinct. P doping is beneficial to the carbonization of N-doped carbon during thermal treatment, which accordingly improves the electrical conductivity of the N-doped carbon. Thus, Pasala et al. [105] modified GF with microcrystalline cellulose and (NH_4_)_2_HPO_4_ and showed the synergetic effect of N and P functional groups toward the performance of the modified GF. 

As surface moieties with high oxygen content, such as –OH groups, as well as –COOH and phosphate groups, were confirmed to enhance the reactivity of vanadium redox reactions, it is reasonable to assume that the co-doping of both amine- and sulfur-containing groups to carbon-based electrodes should also offer high-quality reaction sites due to the high oxygen content and deriving from the same main group. Hence, N and S, as mentioned above, are effective in improving the performance of carbonous felt, and the different outer electron structures between them may result in a synergism. 

To obtain clearer knowledge about the performance of heteroatom functional groups to modify carbonous felt, we list the cell performance and corresponding current density of the electrodes mentioned in the nonmetal doping research in Table 3. The specific size of the electrode and half-cell and the modified electrode used are also listed.

Heteroatom doping leads to charge-separated sites in the carbon materials, with the incorporation of the electrochemical species present in the electrolyte [115]. Therefore, it is an effective approach to enhance the electrochemical properties of carbon materials by regulating the surface electronic structure and generating catalytic sites in VRFBs [116]. Heteroatom doping is the main avenue for morphology-preserved modification, preferably averting the aforementioned drawbacks, such as high costs and unsatisfactory cycle stability. Much progress based on nitrogen, chalcogen, and halogen doping has been achieved in numerous previous works. However, the battery durability is far from desirable. Herein, it is still urgently necessary to develop an ultra-stable electrode with high efficiency.

### 2.4. Carbonous Nanomaterial Modification

Growing carbonous nanomaterials on the electrode is another typical approach to morphology modification. Functional carbon-based materials with diverse properties show great application prospects for the morphology modification of electrode materials in VRFBs, such as CNTs, CNFs, graphene-based materials, and biomass porous carbon. With a high specific surface area and good conductivity, carbonous felt nanomaterials are attractive as electrocatalysts. As the reaction activity sites are on the surface of the carbonous fiber, zero-dimensional (0D) carbonous nanomaterials such as CNPs [117], mesoporous carbon (MPC) [118], and charcoal [119] are easily decorated onto it, aiming to obtain the maximum specific surface area. For example, Wei et al. [117] treated commercial CNPs with nitric acid to increase the active sites, and then decorated it onto GF with the help of Nafion via simple dissolving and drying. The GF with a uniform CNP coating showed EE of 84.8% at 100 mA·cm^−2^.

As mentioned above, mesoporosity most likely improves the transportation of the active species but requires an optimal pore size range to create an accessible and even active surface. Besides CNPs, MPC and charcoal, with abundant holes on the surface, are also used to enlarge the specific surface areas of substrate fibers. For instance, Schmidt et al. [118] generated MPC via a sol–gel process and subsequent carbonation, and they found that the improvement in the surface area increased the active sites and ameliorated the diffusion of vanadium ions. Activated charcoal is widely employed as an electrode material in energy storage devices due to its large surface area, excellent adsorption properties, abundant active sites, and low cost [120]. The in-situ-formed pyrolytic carbon may naturally achieve fine adhesion and low contact resistance, as activated charcoal and GF are carbon materials in essence. To coat GF with evenly immobilized activated charcoal particles, Yang et al. [119] employed a sucrose pyrolysis process, and the surface area of the modified GF was twice that of the pristine GF.

Indeed, 0D carbon-based materials such as CNPs and carbon dots (CDs) [121] attract attention due to their low cost and high electronic conductivity and catalytic activity, resulting from the large specific surface area and plentiful surface defects, effectively enhancing the rate performance as well as electrolyte utilization and power density in the VRFB. Nevertheless, it is challenging to control the dispersion of 0D materials and active materials might suffer from exfoliation from the electrodes during the long-term cycling process. Herein, many researchers have applied one-dimensional (1D) carbonous nano-materials, such as vapor-grown carbon fibers (VGCFs) [122], CNTs [109,123], and CNFs [123]. These catalysts have high specific surface areas and abundant functional groups that provide them with prominent electrochemical catalytic activity.

For example, Yang et al. [122] generated VGCFs with PAN-based oxidized fibers via thermal treatment, and grew nanotubes 20~80 nm in diameter with the catalysis of iron, nickel, or cobalt. The charge–discharge efficiency of the cell with co-doped GF was raised by more than 12%. CVD is an efficient technique for the growth of CNTs on CF. Thus, Wang et al. [109] grew nitrogen-doped CNTs (N-CNTs) on GF with ethylenediamine by CVD at 800 °C in inert gas, and then heated them at 400 °C for 1 h in air. The EE was increased by 8.8% at 10 mA·cm^−2^ over the pristine one.

Meanwhile, CNFs appear to be a promising candidate, ascribed to their specific electrical, physicochemical, and mechanical properties. The graphene layer in the CNF is tilted against the fiber axis, resulting in exposed edge planes toward the exterior surface, together with plentiful active sites for ionic adsorption and chemical bonding directly with the reactants [124]. On account of their unique characteristics and structure, both CNFs and CNTs have become commercial products. The concept of CF electrodes with CVD-grown CNFs and CNTs was further explored by Park et al. [123], with highly enhanced reactivity in both the anolyte and the catholyte of VRFBs. They coated CNFs and CNTs on CF via the pyrolysis of acetylene, with the catalysis of nickel at 700 °C to provide more active sites for the adsorption of vanadium species on the surface of the GC electrode. The discharge capacity and EE of the CNF-and-CNT-co-doped CF increased by about 25% at 100 mA·cm^−2^, respectively, ascribed to the additional surface defect sites of the exposure edge plane in the CNFs and the accelerated charge transfer rate of the in-plane side wall of the CNTs (Figure 11). CNFs and CNTs show great potential as catalysts for CF but the current density still needs to be improved to satisfy the demand for industrial applications.

To improve the performance of carbonous electrodes, the combination of acid or thermal treatment mentioned above with the loading of multi-walled CNTs (MWCNTs) has been used to modify the CF electrode to increase the oxygen functional groups or surface area by Jelyani et al. [125]. Then, the charge transfer resistance of the modified CF was also decreased.

The combination of 0D or 1D carbonous nanomaterials with carbonous felt is too weak, as the contact area is not sufficient and the nanomaterials are also unconnected. Against this background, the deposition of two-dimensional (2D) carbonous nanomaterials such as graphene materials on carbonous felt substrates has emerged as an economic and versatile processing technique [126]. They possess high electrical conductivity, abundant active groups, excellent mechanical stability, feasible electrochemical activity, an especially large specific surface area, and a high surface-to-volume ratio, which could resolve the contact problem associated with 0D or 1D carbonous nanomaterials [127]. Graphene materials, such as GO [128,129,130,131] and carbon sheets (CSs) [132], used to modify carbonous felts would be an effective strategy to promote charge transfer and ion diffusion on the electrode surface effectively, as well as reduce the contact resistance and interfacial polarization for the vanadium ion redox reaction. Different methods such as dip-coating, constant potential techniques, and electrophoretic deposition (EPD) have been used individually or together for the coating of graphene-based materials on felt electrodes.

Gonzalez et al. [129] coated commercial GO onto GF by EPD to efficiently obtain films from suspensions. Then, a 3D cross-linked structure with a unique morphology consisting of fibers interconnected by graphene-like sheets was built and positively contributed to the formation of a conducting network, as well as speeding up the charge transfer kinetics over the pristine GF. Consequently, it reached a high EE value of 95.8% at 25 mA·cm^−2^. Graphene-based materials consisting of sp^2^-hybridized carbon atoms with ample oxygen defects, favoring mass-transport processes and electron transfer, are promising candidates for electrocatalysts in VRFBs, yet they can only enhance the electrochemical activity of half-cell redox couples operating at relatively low current densities (≤50 mA·cm^−2^). The catalytic activity of GO can be improved by different reduction processes, aiming to obtain the high electrical conductivity of graphene. As mentioned above, the hydrothermal treatment is suitable to form uniform nanoparticles. Therefore, Deng et al. [130] modified GO on GF via the hydrothermal method to fabricate a reduced GO-modified GF (rGO-GF) composite electrode with abundant oxygen groups that could accelerate the redox reaction on the surfaces of rGO-GF electrodes. It can work under high current densities (150 mA·cm^−2^), with an outstanding capacity and EE (as shown in Figure 12).

Despite the satisfying results obtained from the application of 2D nanomaterials in VRFBs, there are several shortcomings that must be addressed, such as irreversible stacking and the agglomeration of nanosheets due to strong π–π interaction, resulting in decreased active surface areas. During the preparation process of composite electrodes, the GO or rGO with a 2D structure is inclined to exhibit the curling and agglomeration problem, with difficulty in achieving a large specific surface, and this also restricts the electronic conduction and vanadium ion diffusion throughout the electrode material, which largely influences the rate capability and EE of the VRFBs. The fabrication of 3D nano-materials, such as CNS [133,134], carbon microspheres (CMS) [135], and 3D wrinkled mesoporous graphene (MG) [136], could be an effective strategy to overcome the limitations of 2D graphene. Many researchers have attempted to transform 2D graphene nanosheets into 3D porous structures via various techniques, such as hydrothermal processes, aerosol spray drying, and thermal reduction methods [137]. In addition to retaining the inherent physiochemical properties of 2D graphene nanosheets, they gain a 3D conductive network, mesoporous and microporous structures, and ultra-high specific surface areas, making them promising materials for energy storage devices.

It has been corroborated that dopamine monomers can be oxidized with the presence of oxygen and conveniently self-polymerize under alkaline environments, requiring neither complicated instruments nor toxic chemicals. Therefore, Wu et al. [133] employed dopamine as a carbon and nitrogen source simultaneously to generate N-doped CNS on GF via a self-polymerization procedure followed by carbonization. The EE of the cell with N-doped CNS-modified GF was 72.2% and 53% at 150 mA·cm^−2^ and 300 mA·cm^−2^, respectively. Furthermore, the cell not only retained prominent durability in long period cycles at 150 mA·cm^−2^, but also displayed notable adjustability in a wide temperature range from 15 to 50 °C, with high energy efficiency. The better electrochemical properties of NCS/GF may be ascribed to the increase in N-containing groups, which create extra active sites to absorb the vanadium ions easily and accelerate ion exchange during the charge–discharge process. Moreover, He et al. [135] coated GF with glucose to form CMS via the hydrothermal method and ground it with (NH_4_)_2_HPO_4_ to introduce N and P functional groups. The EE of the modified GF was increased by 6.5% over the original GF at 100 mA·cm^−2^, which may be derived from the increase in the total number of heteroatoms as active sites and the synergistic effect of N and P co-doping due to the formation of the unique structural features of CMS. However, the performance of the modified electrode is not superior to that demonstrated in other studies. Nevertheless, the hydrothermal method is able to form uniform CMS and is worth further application.

As the whole carbon sphere must be attached to the GF, the contact area needs to be increased. Therefore, Zhao et al. [134] synthesized a nano-micro hierarchical electrode by coating CNS on GF (CNS-GF), employing SiO_2_ nanospheres as templates and phenol formaldehyde resin (PF) as a carbon source (Figure 13a). The formation of a carbon layer ensured the shape of both SiO_2_ and the graphite fibers, thus raising the surface area of the graphite fibers. The carbon layer offered adequate stable active sites, and it was also able to endure the corrosion and washout of the electrolyte. It showed stable EE of 75.5% at 150 mA·cm^−2^ even after 5000 cycles over 3000 h, exhibiting strong durability at a high current density during ultra-long-term charge–discharge cycles, as shown in Figure 13d.

Furthermore, to load sufficient catalyst, Opar et al. [136] immersed and dried the GF into a mixture of GO and a triblock copolymer several times to complete a self-assembly interaction, followed by carbonation to form MPG. Owing to the plentiful 3D wrinkled mesoporous structures, enlarged specific surface area and pore volume, improved wettability, and electrical conductivity, the obtained 3D MG-modified CF (MG-CF) showed EE of 76.5% and 61.6% at 100 mA·cm^−2^ and 175 mA·cm^−2^, respectively.

To realize low carbon emissions, biological precursors are more attractive due to their low price and extensive sources [138]. Therefore, corn protein [139,140], sugarcane bagasse (SCB) [141], shaddock peel [142], fungus [143], scaphium scaphigerum [144,145], kiwi fruit [146], lotus seed shells [147], and silk-protein-derived carbon fabric [148] have been chosen as biomass carbon sources to create porous carbon catalysts. Park et al. [139] grew a layer of corn protein on the CB and then heated it at 800 °C. The corn protein also served as an N source and the N-doped CB was decorated on the CF by dipping and drying. The EE of the cell with N-doped-CB-modified CF was 85.2% after 100 stable cycles at 50 mA·cm^−2^. Similarly, Aziz et al. [140] also chose corn protein to coat carbon nanorods (CNRs), which was performed via electrospinning with PAN. After being heated at 900 °C for 3 h in Ar, the N-doped CNRs were decorated onto GF by simple dipping and drying. The EE of the cell with N-doped-CNR-modified CF (N-CNRs-CF) was 84.3% at 40 mA·cm^−2^. As the performance of the N-CNRs-CF was close to that of the N-CB-CF, the superiority of the electrospinning method was not obvious and the method required additional electrospinning instrument, which increased the cost.

Moreover, Mahanta et al. [141] treated SCB with KOH and then heated it at 230 °C for 12 h and at 700 °C for 2 h under N_2_; they then decorated it onto GF using dipping and drying procedures. The electrochemical surface area increased by 80 times and a higher current was observed due to a nearly two-times enhancement in the electrochemical active surface area of T-GF/AC-SCB. At a current density of 100 mA·cm^−2^ after 50 cycles, the carbon-coated electrode exhibited EE and VE of 72% and 77%, respectively, which was higher than that of the thermally treated GF. Sugarcane bagasse shows potential in modifying carbonous felt, but other biological wastes should also be investigated to identify more effective precursors. Meanwhile, their stability and pretreatment should also be taken into consideration, in addition to the degradation of the battery lifetime. However, the wide application of biomass waste has been prohibited by the tedious preparation and rigorous reaction conditions.

In Table 4, it is clear that the carbon materials doped on carbonous felt electrodes are purchased or carbonated from organics previously coated on electrodes. However, when surface functional groups are formed through the above chemical or electrochemical oxidation reactions on the graphitic surfaces, the graphitic sp^2^ bonds are interrupted to transform into more stable and non-conductive sp^3^ bonds, which counteracts the electron transfer of the redox reaction process. As a result, the electron transfer through these surface functional groups is hindered, resulting in sluggish redox reactions and failure to meet the demands of VRFB applications. Thus, the conventionally recognized reaction mechanisms involving generated surface functional groups as a paradigm are highly disputable. These functional groups might simply serve as wetting agents by absorbing water molecules, e.g., through hydrogen bonding, to the electrode surface. For a better understanding of the four different modification techniques mentioned above, the main advantages and drawbacks of each approach are summarized in Table 5.

## 3. Conclusions and Perspectives

Carbonous felts including CF and GF have been extensively employed as potential electrode materials in electrochemical energy storage devices, due to their good chemical stability, electronic conductivity, portability, and low cost. However, untreated carbon-based electrodes have poor catalytic activity for redox reactions and cannot satisfy the development demands of VRFBs. Therefore, it is crucial to modify carbon-based materials to improve their electrochemical performance and reduce the polarization loss. To date, most studies have focused on enhancing the electrocatalytic activity of the electrode materials to reduce the electrochemical polarization, by modifying the surface functional groups, adjusting the microstructure of the carbon material, increasing the surface area, and loading electrocatalysts. These interesting results offer new insights for the further development of reliable VRFB systems. Some main conclusions can be drawn from our comprehensive analysis of the studies, divided into four aspects, as follows:Traditional surface treatments as well as novel intrinsic treatments all belong to the category of morphology-retaining methods, transforming the hydrophobic surfaces of bare felts into hydrophilic ones by the attaching various oxygen functional groups to the surfaces of the electrodes. The enhanced wettability makes it more feasible for the vanadium ion to migrate and obtain access to the active sites of the modified electrodes. Certain –OH or –COOH functional groups are widely regarded as offering active reaction sites for the vanadium redox reaction, enhancing the reactivity and wettability of the carbon electrodes. However, these new surface modification methods rely on the development of specific equipment or complex operations, restricting their large-scale application.Metallic or metal oxide nanoparticles attached to the carbon-based electrode via modification significantly improve the electrical conductivity and the electro-active surface, which ensures low polarization and enhanced reversibility in vanadium redox reactions. In particular, to address the issue of HER on the precious metal catalysts, various metal oxides have been presented to abate the overpotential in the negative half-cell. However, the functional and exact mechanisms of metal oxides or their superiority over others in most of the relevant studies remain ambiguous. The questions of how to ensure uniformity of the coating and reduce the costs of integrating oxygen functional groups on the surfaces of electrodes are worth intensive investigation. Furthermore, it must be noted that the deposited electrocatalyst can be exfoliated from the carbonous felt electrode under the flushing of the circulated acid electrolyte, degrading the electrode’s cycling stability and durability. Notably, novel nanosized metallic alloys with good corrosion resistance, a high surface area, and excellent electrical conductivity can be proposed as a possible candidate to generate additional catalytic activity.Heteroatom doping, including O, P, PO_3_, F, and SO_3_, is an effective approach that introduces functional groups into the matrix, regulating the surface electronic structure and generating catalytic sites and functional groups. With respect to heteroatom doping, diverse dopants such as P, B, and S, etc., have been also shown to display excellent electrochemical performance in VRFB applications. In particular, N groups are very attractive since they can synchronously improve the hydrophilicity and electrocatalytic activity, as well as the electrical conductivity. Significantly, dual- or even triple-doped carbon nanomaterials, such as B- or N-doped graphene and P-, S-, or N-doped carbon, easily controlling the electronic states of carbon materials, have been considered and will represent the predominant development direction in the future. Nonetheless, a more efficient and environmentally friendly heteroatom doping method is essential for industrial production.The surface area of the electrode could be enlarged by etching or decorating well-tailored carbon-based nanomaterials of different dimensions on the matrix of the electrode, which can provide a large surface area, rich surface functional groups, and superb electrical conductivity comparable to that of a metallic catalyst. Most surface moieties could serve as dominant active sites for redox reactions and promote the adsorption and desorption processes of vanadium ions, resulting in accelerated V^2+^/V^3+^ and VO^2+^/VO_2_^+^ reaction kinetics and high mass transport kinetics. Patterned microporous electrodes possessing a specific surface area and permeability would be promising. Researchers are expected to devote more efforts to developing novel functional carbon materials based on CNTs to promote the V^2+^/V^3+^ and VO^2+^/VO_2_^+^ redox reaction kinetics, although achieving their application without any performance deterioration will be a major challenge regarding long-term operation in practical VRFB systems.

Currently, the combination of the mentioned four types of modification methods tends to attract much attention, particularly for methods such as the decoration of both metal oxide particles and graphene, polymers with graphene, or CNTs. Therefore, the choice of a reasonable method to improve the kinetics of redox reactions and the hydrophilicity of carbon-based electrodes will still be an important topic in the future studies. Independent electrode materials suitable for both positive and negative electrolyte reaction mechanisms need to be further explored. Moreover, the pursuit of good corrosion-resistant carbon substrates during overcharge with low costs will secure the longevity of VRFB systems. Carbonous felts will remain the dominant materials for electrodes for VRFBs at present, but a future challenge will be to employ a thin carbon cloth or paper as part of a “zero-gap” cell architecture so as to reduce the ohmic polarization of VFBs to a large degree and obtain a high-power-density VRFB stack.

On the whole, the high cost of VRFBs is the main issue that hinders their practical application in large-scale demonstrations. Biomass precursors and facile methods have taken precedence to reduce the costs. More thorough investigations will be indispensable to accurately grasp the essential vanadium redox reaction’s mechanism arising on the electrode surface. Moreover, in-depth and more extensive research that includes cell design along with the effects of the main electrolyte on the overall battery performance should also be addressed. In summary, to improve the performance of VRFBs and improve their commercial application, more profound and comprehensive research should be carried out. As the requirement for energy storage systems with high power density continues to rise, the potential mechanism of the vanadium redox reaction with regard to different electrocatalysts should also be elaborated so that VRFBs can undergo further development and achieve wider application.

## Figures and Tables

**Figure 1 materials-16-03811-f001:**
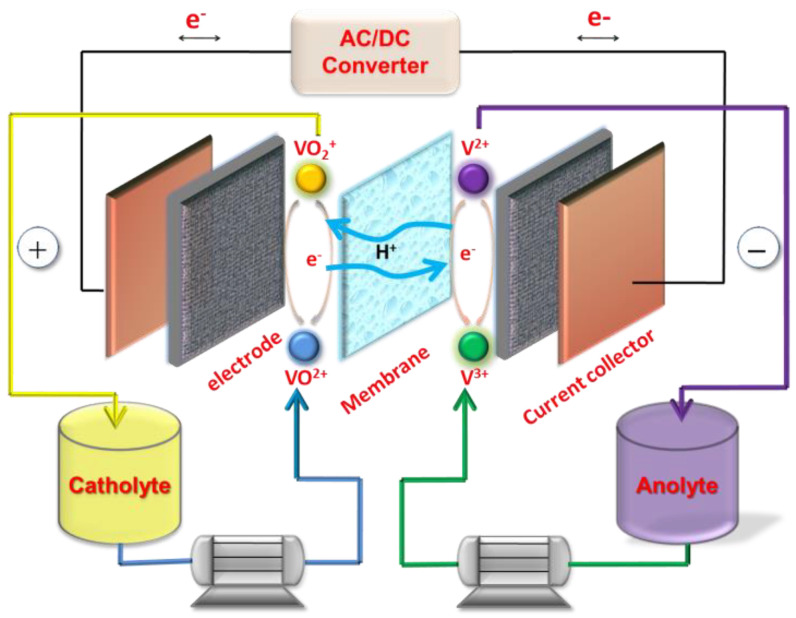
Schematic diagram of the vanadium flow battery composed of key materials. Copyright 2013 by American Chemical Society [3].

**Figure 2 materials-16-03811-f002:**
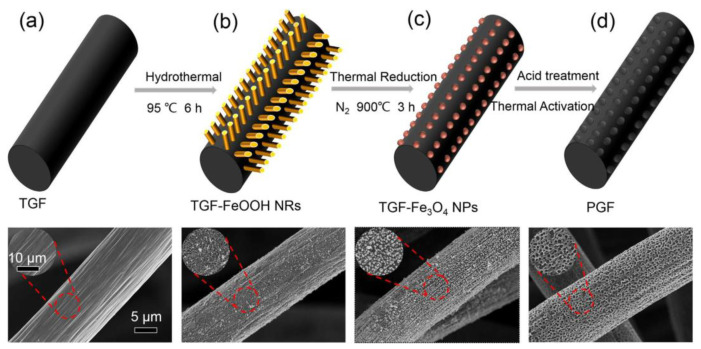
Illustration of the porous graphite felt (PGF) preparation procedure and the corresponding SEM images of each step: (**a**) pristine GF after thermal treatment; (**b**) FeOOH nanorods grown on TGF via hydrothermal method; (**c**) TGF-Fe_3_O_4_ NPs by thermal reduction; (**d**) the holey-engineered PGF obtained by acid treatment. Copyright 2018 by Elsevier Ltd. All rights reserved [23].

**Figure 3 materials-16-03811-f003:**
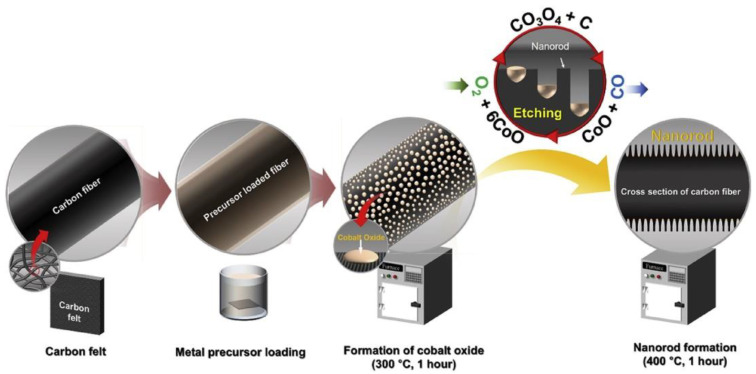
A schematic view of the etching and nanorod formation process on the surface of CF. Copyright 2018 by Elsevier Ltd. All rights reserved [31].

**Figure 4 materials-16-03811-f004:**
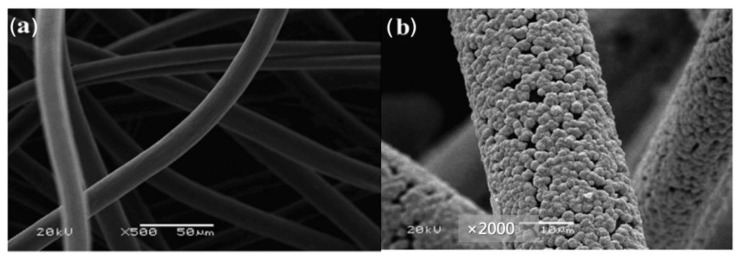
SEM photographs of GF. (**a**) Bare GF and (**b**) GF modified with PbO_2_ through pulse electrodeposition. Copyright 2014 by Elsevier Ltd. All rights reserved [52].

**Figure 5 materials-16-03811-f005:**
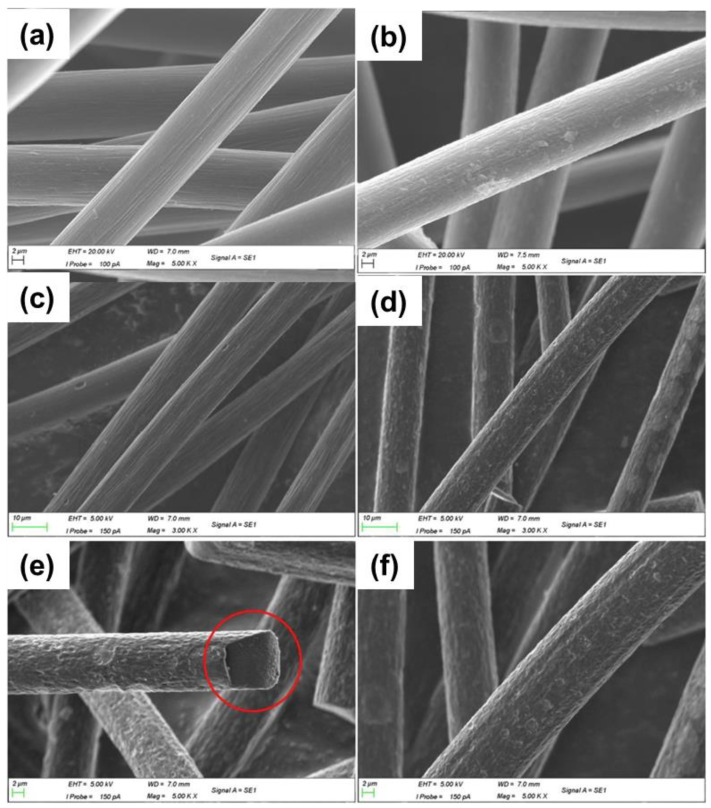
SEM images of (**a**) GF, (**b**) CrTGF (0.2 g), (**c**) GF × 3000, (**d**) CoTGF × 3000, (**e**,**f**) CoTGF at larger magnification ×5000. Copyright 2019 by Elsevier Ltd. All rights reserved [56,57].

**Figure 6 materials-16-03811-f006:**
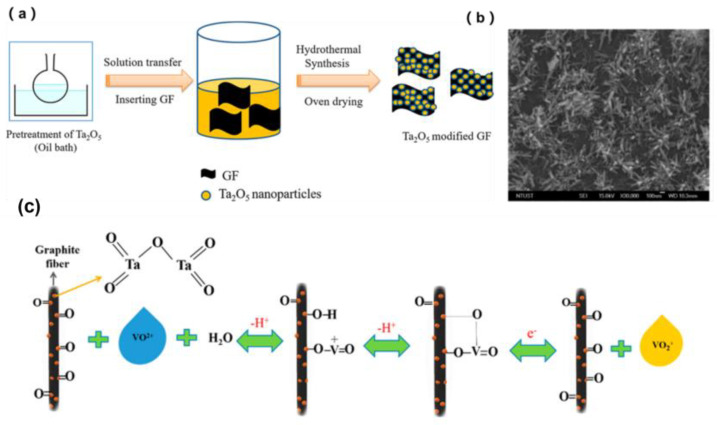
(**a**) Schematic representation of the synthesis of Ta_2_O_5_-GF, (**b**) SEM images of 0.75 wt % Ta_2_O_5_-GF, and (**c**) schematic Illustration of the mechanism of the VO^2+^/VO_2_^+^ redox reaction occurring in the presence of Ta_2_O_5_ nanoparticles on the surface of the GF electrode. Copyright 2018 by American Chemical Society [45].

**Figure 7 materials-16-03811-f007:**
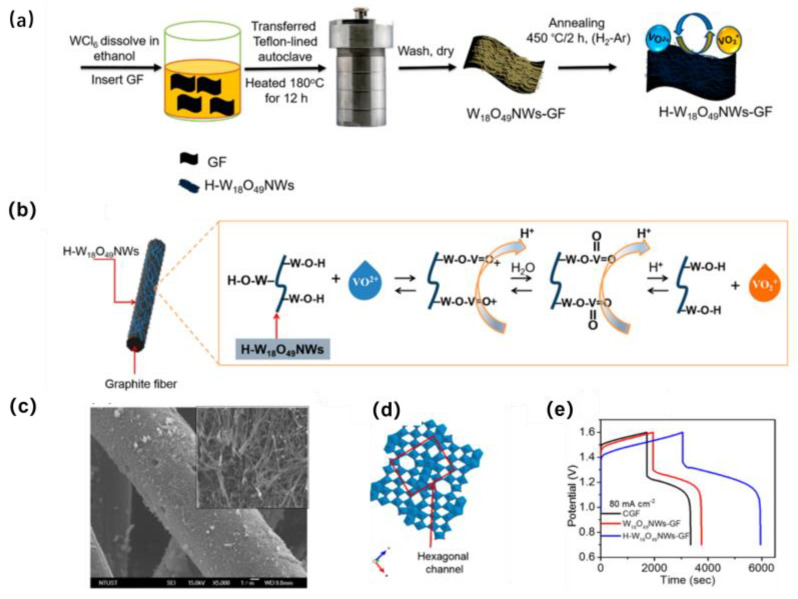
(**a**) Schematic showing the formation of W_18_O_49_NW-GF and H-W_18_O_49_NW-GF electrodes; (**b**) scheme of the proposed catalytic reaction mechanisms for the redox reaction toward VO^2+^/VO_2_^+^ using W_18_O_49_NWs to modify the GF surface; (**c**) SEM image of W_18_O_49_NW with 4.5 mg cm^−2^ catalyst loading on the surface of GF; (**d**) crystalline structure of W_18_O_49_; (**e**) charge−discharge curves of CGF, W_18_O_49_NW-GF, and H-W_18_O_49_NW-GF at the current density of 80 mA·cm^−2^. Copyright 2019 by American Chemical Society [63].

**Figure 8 materials-16-03811-f008:**
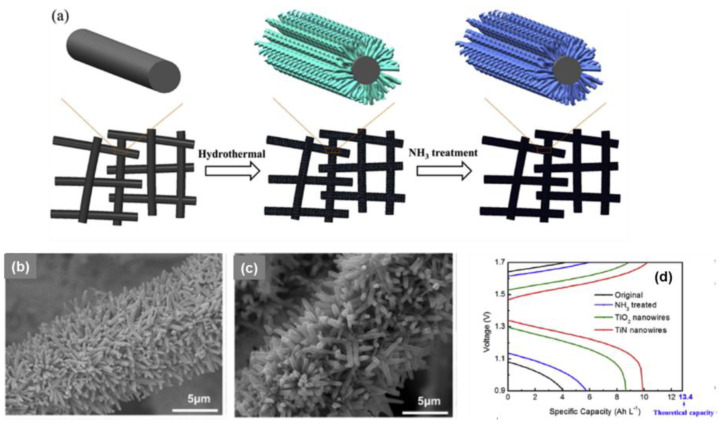
(**a**) Schematic diagram of the two-step growth process to prepare TiN nanowires on the surface of graphite felt; SEM images collected for TiO_2_-decorated electrode; (**b**,**c**) TiO_2_ nanowires annealed in NH_3_ at temperature of 800 °C; (**d**) charge and discharge curve at 300 mA·cm^−2^ of VRFBs equipped with pristine, NH_3_-treated, TiO_2_-decorated, and TiN nanowire (nitrided at 800 °C) array-decorated graphite felt electrodes. Copyright 2017 by Elsevier Ltd. All rights reserved [66].

**Figure 9 materials-16-03811-f009:**
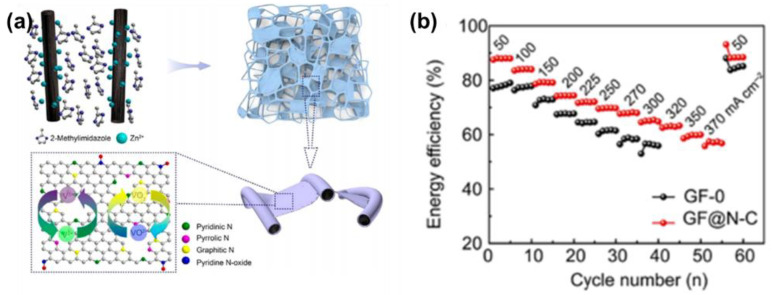
(**a**) Schematic illustration of the preparation of the robust GF-CN-2 electrode and its application in the VRFB and (**b**) the EE of the VRFB based on GF and GF@N-C electrodes at different current densities. Copyright 2018 by American Chemical Society [93].

**Figure 10 materials-16-03811-f010:**
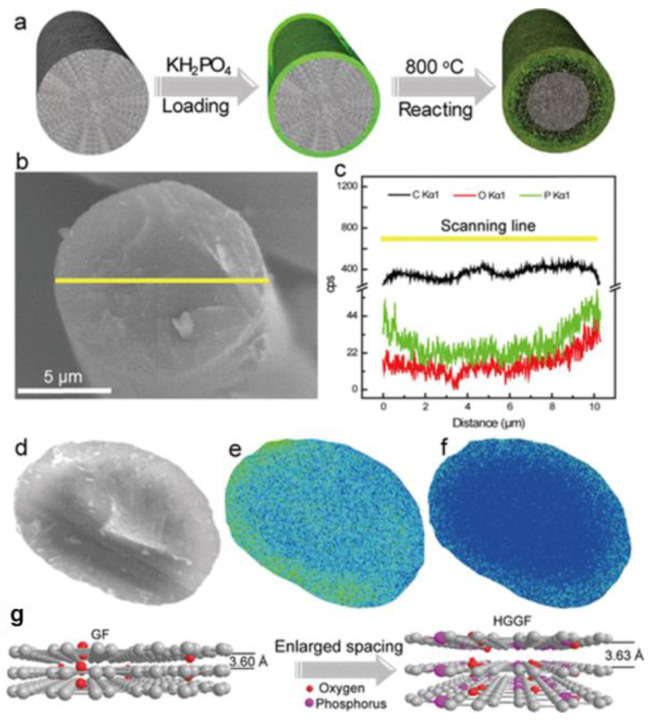
(**a**) Schematic diagram of the fabrication process of the heteroatom gradient graphite felt (HGGF) electrode; (**b**) SEM image of HGGF and (**c**) the corresponding EDS signals of C (black), O (red), and P (green) elements; (**d**) EPMA of the cross-section of the HGGF electrode and (**e**,**f**) corresponding element mapping of (**e**) O and (**f**) P; (**g**) the change in the carbon layer of HGGF. Copyright 2019 by American Chemical Society [103].

**Figure 11 materials-16-03811-f011:**
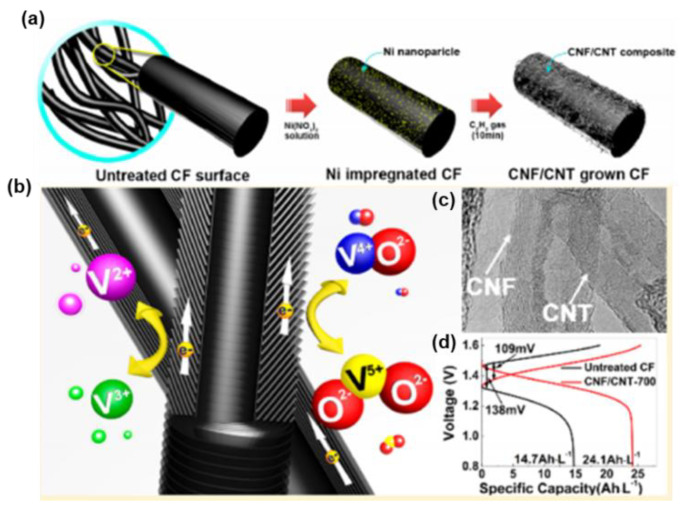
(**a**) Schematic view of the synthesis of CNF/CNT grown on carbon felt surface on Ni nanoparticle seeds via C_2_H_2_ gas decomposition above 500 °C; (**b**) schematic of the mechanism of the vanadium redox reaction on the CNF/CNT-T electrode; (**c**) HR-TEM images of the coexistence structure in CNF/CNT-700 sample and (**d**) charge−discharge voltage profiles of untreated and as-prepared electrodes at 40 mA·cm^−2^. Copyright 2013 by American Chemical Society [123].

**Figure 12 materials-16-03811-f012:**
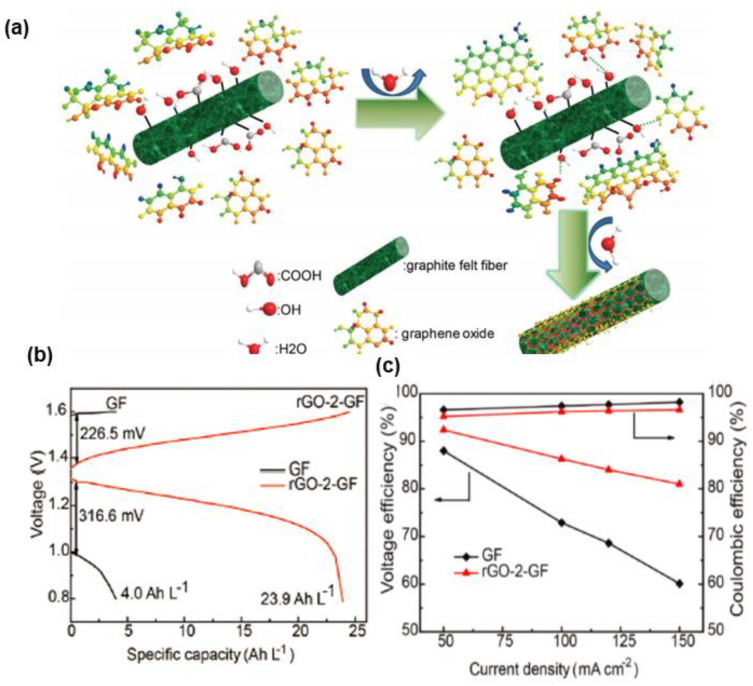
(**a**) Schematic diagram of the fabrication mechanism of the rGO-GF electrode; (**b**) electrochemical performance of a VRFB employing GF and rGO-2-GF electrodes in single cells (**a**) at 150 mA·cm^−2^ and (**c**) VE, CE at different current densities (50, 100, 120, 150 mA·cm^−2^). Copyright 2017 by WILEY-VCH Verlag GmbH & Co. KGaA, Weinheim, Germany [130].

**Figure 13 materials-16-03811-f013:**
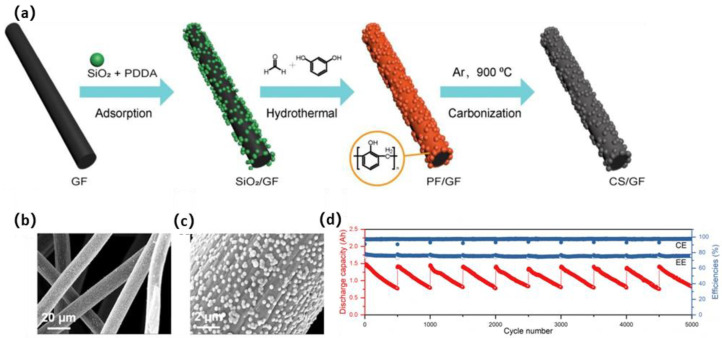
(**a**) Fabrication procedure of the CS/GF hierarchical electrode; SEM images of (**b**) GF, (**c**) CS/GF, and (**d**) ultra-long-term cycling performance of VRFBs with CS/GF electrodes over 10 rounds of 500 cycles test at a current density of 150 mA·cm^−2^. Copyright 2018 by Elsevier Ltd. All rights reserved [134].

**Table 1 materials-16-03811-t001:** Modification methods and reagents used in the literature.

Modification Method	Modification Reagent	Electrode Size	Electrode	Current Density/mA·cm^−2^	EE	Ref.
Microwave	Ar and O_2_	3 × 3 cm^2^	P and N	40	85%	[14]
Square wave potential pulse	NaOH	3 × 3 × 0.6 cm^3^	/	30	87.0%	[15]
Hydrothermal method	HF/H_2_O_2_	3 × 3 cm^2^	P	50	75.7%	[17]
Heating	Air	5.4 × 6 × 0.3 cm^3^	/	40	75%	[18]
Heating	CO_2_	5 × 5 cm^2^	/	50(80)	84.15% (~78%)	[20]
Heating	Water vapor	5 × 5 × 0.65 cm^3^	P and N	80(50)	78.12% (83.10%)	[21]
Heating and etching	Ni(NO_3_)_2_·6H_2_O HNO_3_ and HCl	16 × 0.3 cm^3^	/	150	~71% (VE = 72.8%)	[22]
Hydrothermal method and heating and etching	N_2_/ HCl	5 × 5 × 0.5 cm^3^	/	300(50)	57.3% (~75%)	[23]
Heating	C_4_H_6_O_4_·Co·4H_2_O/Air	25 cm^2^	/	50	87.3%	[31]
Heating and etching	FeCl_3_/Air/HCl	2 × 2 cm^2^	/	200 (500)	87.02% (73.23%)	[30]
Heating	K_2_FeO_4_	2 × 2 cm^2^	/	200 (500)	79.74% (63.41%)	[34]
Vapor treatment	HNO_3_	2 × 2 cm^2^ × 4.2 mm	P and N	250	71.86%	[35]

**Table 2 materials-16-03811-t002:** A list of precursors used to modify carbonous felt electrodes and EE of cells assembled with electrodes modified with metals or metal oxides or their derivatives.

Precursor	Electrode Modified	Electrode Size	Electrode	Current Density/mA·cm^−2^	EE	Ref.
CuSO_4_	Cu-GF	4.7 × 0.15 cm^3^	/	300 (200)	80.1% (84%)	[39]
α-PbO_2_ and β-PbO_2_	PbO_2_-GF	3 × 4 × 0.3 cm^3^	P	80	78.1%	[52]
Cr(NO_3_)_3_	Cr_2_O_3_-GF	2 × 2 × 0.3 cm^3^	P and N	150	67.6%	[56]
Co(NO_3_)_2_	CoO-GF	2 × 2 × 0.3 cm^3^	P and N	150	69.4%	[57]
Ni(NO_3_)_2_	NiO-GF	16 × 0.3 cm^3^	P and N	125	74.5%	[58]
Ta_2_O_5_	Ta_2_O_5_-GF	5 × 5 cm^2^	P and N	80	73.73%	[45]
SnCl_4_	SnO_2_-CF	5 × 5 cm^2^	/	150	77.3%	[46]
WCl_6_	H-W_18_O_49_NWs-GF	5 × 5 × 0.5 cm^3^	/	160	66.1%	[63]
Na_2_WO_4_ and NbCl_5_	Nb/h-WO_3_ NWs-GF	5 cm^2^	/	160	65.83%	[71]
Niobium ethoxide and titanium isopropoxide	TiNb_2_O_7_-GF	5 × 5 × 0.65 cm^3^	/	160	70.32%	[72]
Co(NO_3_)_2_ and NiCl_2_	NiCoO_2_-GF	2 × 2 × 0.3 cm^3^	P and N	150	72.5%	[73]
Titanium butoxide and H_2_	TiO_2_:H-GF	4 × 0.5 cm^3^	N	150	66.1%	[64]
Titanium butoxide and NH_3_	TiO_2_:N-CF	4 × 0.5 cm^3^	N	150	71%	[65]
TiCl_4_ and NH_3_	TiN-GF	4.7 × 0.15 cm^3^	N	300 (200)	77.4% (82.8%)	[66]
Ti_3_C_2_T_x_	Ti_3_C_2_T_x_-GF	1.95 × 3 × 0.6 cm^3^	N	200 (300)	81.3% (75%)	[70]
TiF_4_	TiC-GF	5 × 4 cm^2^	N	100	~74%	[67]
C_4_H_6_CoO_4_⋅4H_2_O	Co_3_O_4_-CF	3.3 × 3.3 cm^2^	P and N	160 300	71.4% (55.8%)	[74]

**Table 3 materials-16-03811-t003:** Sources of heteroatoms used to prepare modified carbonous felt electrodes doped with nonmetals and corresponding EE of cells at different current densities.

Source of Heteroatom	Electrode Modified	Electrode Size	Side Modified Electrode Used	Current Density/mA·cm^−2^	EE	Ref.
Pyrrole	N-GF	3 × 3 × 0.3 cm^3^	/	150	74.2%	[77]
Dopamine	N-GF	3 × 3 cm^2^	P and N	150	75.8%	[78]
Dopamine	N-GF	3.3 × 3.3 × 0.42 cm^3^	/	150	75.5%	[79]
Urea	N/C-GF	2 × 2 cm^2^	/	200	64.1%	[92]
Zn(NO_3_)_2_ and 2-methylimidazole	N/C-GF	2 × 2 × 20.5 cm^3^	/	200 (370)	74.3% (56.9%)	[93]
p-Phenylene-diamine and phytic acid	N/C-GF	4 cm^2^	/	200	65.4%	[94]
PIM	N/C-GF	3 × 3 cm^2^	/	150	~71.5%	[95]
NH_3_ and O_2_	N/O-GF	5 × 3 × 0.5 cm^3^	P and N	110	~73%	[97]
Urea; NH_3_ and O_2_	N/O-GF	5 × 3 × 0.5 cm^3^	P and N	80	~71%	[98]
Urea	N-GF	5 × 8 × 0.6 cm^3^	/	150	81.32%	[83]
N_2_	N-GF	3.2 × 3.2 × 0.6 cm^3^	P and N	80	76.8%	[80]
NH_3_	N-CF	4 × 5cm^2^	P	100	85%	[82]
N_2_ plasma	N-CF	5 × 5cm^2^	/	64	~70%	[81]
HEDP	P-GF	5 × 5cm^2^	/	150 (400)	~79% (~52%)	[87]
NH_4_PF_6_	P-O-CF	5 × 5cm^2^	P and N	120	~75.5%	[102]
KH_2_PO_4_	HGGF	/	/	150 (300)	73.34% (60%)	[103]
Microcrystalline cellulose and (NH_4_)_2_HPO_4_	N-P-GF	16 cm^2^	N (cathode)	70	~74%	[105]

**Table 4 materials-16-03811-t004:** Precursors used to modify carbonous felt nanomaterials on carbonous felt electrodes and EE of cells obtained with them at different current densities.

Precursors	Electrode Modified	Electrode Size	Side Modified Electrode Used	Current Density /mA·cm^−2^	EE	Ref.
CNP	N-CNP-GF	1.8 × 2.6 × 0.6 cm^3^	/	200	72.7%	[117]
Sucrose	Charcoal-GF	3 × 3 × 0.3 cm^3^	P	100	81.7%	[119]
Ethylenediamine	N-CNT-GF	5 cm^2^	/	10	77%	[109]
C_2_H_2_	CNF/CNT-CF	5 cm^2^	/	100	65.6%	[123]
GO	G-GF	4 cm^2^	P	25	95.8%	[129]
GO	rGO-GF	2 × 2 × 0.5cm^3^	/	300	60%	[130]
Phytic acid and urea	GF@CS	2 × 2cm^2^	/	150 (300)	74.79% (~54%)	[132]
Dopamine	N-CNS-GF	5 × 5 × 0.5 cm^3^	/	300	53%	[133]
Glucose and (NH_4_)_2_HPO_4_	N/P-CMS-GF	3 × 3 cm^2^	P	100	67.1%	[135]
PF and SiO_2_ nanospheres	CS-GF	5 × 5 × 0.5 cm^3^	/	150 (300)	75.5% (63.1%)	[134]
GO and triblock copolymer	MG-CF	6 × 0.42 cm^3^	/	200	53.3%	[136]
Corn protein and CB	N-CB-CF	5 cm^2^	P and N	150	68.6%	[139]
Corn protein and PAN	N-CNRs-CF	9 × 0.3 cm^3^	P and N	160	~79.5%	[140]
SCB	SCB-GF	5 × 5 × 0.6 cm^3^	/	100	~72%	[141]

**Table 5 materials-16-03811-t005:** Comparison of the four types of modification approaches to improve the performance of pristine carbonous felt electrodes.

Modification Methods	Main Advantages	Main Drawbacks
Surface treatment or structure rebuilding	Increased specific surface area, enriched oxygen functional groups, high electrochemical activity, enhanced wettability, reduced polarization, and prominent energy efficiency.	Corrosion of electrode caused by excessive surface oxidation; oxygen functional groups may not be stable enough during cycling.
Metallic modification	Improved electrical conductivity, lower polarization, and better kinetic reversibility and electrochemical performance.	Highly energy-/time-consuming and environmentally unfriendly, high cost of production, and relatively weak adhesion between the grown metal species and the carbonous matrix.
Nonmetal doping	Improved electron distribution and local bonding environment on the surface of carbon-based materials, hydrophilicity and electrocatalytic activity, as well as improved electrical conductivity; reasonable cost and satisfactory cycle stability.	Unsatisfactory battery durability; more efficient and environmentally friendly heteroatom doping techniques should be explored.
Carbonous nanomaterial modification	Enlarged surface area of electrode, rich surface functional groups, superb electrical conductivity, promotion of adsorption and desorption processes of vanadium ions, accelerated reaction kinetics, and high mass transport kinetics.	0D or 1D carbonous nanomaterials: insufficient contact area and unconnected nanomaterials. 2D carbonous nanomaterials: irreversible stacking and the agglomeration of nanosheets due to strong π–π interaction, resulting in decreased active surface area.

## Data Availability

Not applicable.

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
