# Peer review of "Insights into the Modification of Carbonous Felt as an Electrode for Vanadium Redox Flow Batteries"

_materials, 2023, doi:10.3390/ma16103811_

Round 1

Reviewer 1 Report

This paper represents an important subject of carbon electrodes modification in the context of VRFB performance improvement. I consider this manuscript highly relevant and interesting for the auditory of Materials. The manuscript contains all main approaches of electrode modifications with sufficient number of references. The whole structure of review is well organised and conclusions support the main text.  There are a few grammar/punctuation mistakes which can be solved during production. 
I suggest that this paper can be published in Materials in present form.

Reviewer 2 Report

The review “Insights into the modification study of carbonous felt as electrode for vanadium flow batteries” summarized relevant results for carbonous felt-based electrodes and can be considered for publication in Materials after improvement concerning specific comments, as follows:

11)     The language must be revised, as examples:

Line 76-77 Please revise the sentence “And metals applied to VFB are gold, lead, titanium, platinum, titanium based platinum and titanium based iridium oxide etc”

Line 82 – the same for “Carbonaceous materials are various but only some of them are normally

22)     Section 2.3 – Nonmetal: the influence of conducting polymers is poorly explored in this section – the use of pyrrole is cited as a precursor in Table 3. I suggest improving the discussion relative to this class of materials.

TThe language must be revised, as examples:

Line 76-77 Please revise the sentence “And metals applied to VFB are gold, lead, titanium, platinum, titanium based platinum and titanium based iridium oxide etc”

Line 82 – the same for “Carbonaceous materials are various but only some of them are normally

Reviewer 3 Report

The authors presented a very good review paper on the improvement of the characteristics of pristine carbon felt materials with different modifications. Paper cover most of the papers (up to date) dealing with the subject matter. The paper could be accepted after minor revisions.

The reviewer could suggest that authors add as Figure 1, a schematic presentation of vanadium redox flow batteries. 

“vanadium flow batteries”

Reviewer: The usual name is “vanadium REDOX flow batteries”

Abstract

Due to the high cost of reaction irons in vanadium flow batteries

Reviewer: What does it mean “.....of reaction irons.....”

Eqs. 1-3 are too big, letters should be smaller, also “disch”charge,  disch should be in a regular form not italic.

Table 1. The current density should be mA cm^-2, not mA cm-2.

Check the Table 3 caption.

Reviewer 4 Report

In this manuscript, the authors tried to brief on the modification study of carbonous felt as electrode for vanadium flow batteries annexed to the materials physicochemical structures and their aligned electrochemical properties. Moreover, the authors presented important insights on modification methods of carbonous felt to be used as electrode for vanadium flow batteries. However I missed discussion on important challenges related to the topic. In this review, the authors pointed about important strategies to improve the performance of carbonous felt electrode, such enhancing the presence of functional groups and increasing of surface area.

This review is meaningful, with important issues being well addressed, and I would like to recommend its publication after moderate revisions.

The authors stressed that “this review will concentrate on the technical trends in each modification method category, including preparation and decoration of carbon based felt electrodes, structure and morphology characterization, electrochemical or cell performance evaluation for the VFB”…..I suggest exploring in one or two figures the main reactions/methods for the main explored modification techniques. This would help the readers to understand better the paper.

Maybe summarize in a Table the main advantages and drawbacks of the mentioned methods.

It would be very interesting to summarize in sentences the main challenges faced on the proposed review topic.

Adjust the references in the Tables, keep only the number format.

The authors pointed out that “high cost of VFB is the main issue that hinders it from practically applied in demonstration.”…do you have figures to compared with LIBs?

Round 2

Reviewer 2 Report

I consider that manuscript can be accepted after revision provided by the authors. Additional corrections in language is necessary.

I suggest a review by a native previously to the final publication.